# Glucose/Graphene-Based Aerogels for Gas Adsorption and Electric Double Layer Capacitors

**DOI:** 10.3390/polym11010040

**Published:** 2018-12-28

**Authors:** Kang-Kai Liu, Biao Jin, Long-Yue Meng

**Affiliations:** 1Department of Chemistry, Yanbian University, Park Road 977, Yanji 133002, China; lkk0391@163.com (K.-K.L.); jinbiao@ybu.edu.cn (B.J.); 2Department of Polymer Materials and Engineering, Department of Chemistry, MOE Key Laboratory of Natural Resources of the Changbai Mountain and Functional Molecules, Yanbian University, Park Road 977, Yanji 133002, China

**Keywords:** graphene, glucose, aerogels, adsorption, electrochemical performance

## Abstract

In this study, three-dimensional glucose/graphene-based aerogels (G/GAs) were synthesized using the hydrothermal reduction and CO_2_ activation method. Graphene oxide (GO) was used as a matrix, and glucose was used as a binder for the orientation of the GO morphology in an aqueous media. We determined that G/GAs exhibited narrow mesopore size distribution, a high surface area (763 m^2^ g^−1^), and hierarchical macroporous and mesoporous structures. These features contributed to G/GAs being promising adsorbents for the removal of CO_2_ (76.5 mg g^−1^ at 298 K), CH_4_ (16.8 mg g^−1^ at 298 K), and H_2_ (12.1 mg g^−1^ at 77 K). G/GAs presented excellent electrochemical performance, featuring a high specific capacitance of 305.5 F g^−1^ at 1 A g^−1^, and good cyclic stability of 98.5% retention after 10,000 consecutive charge-discharge cycles at 10 A g^−1^. This study provided an efficient approach for preparing graphene aerogels exhibiting hierarchical porosity for gas adsorption and supercapacitors.

## 1. Introduction

Given the excessive consumption of fossil fuel, environmental pollution and the energy crisis have become increasingly prominent. Graphene presents a two-dimensional (2D) macrostructure featuring excellent interfacial properties. Because of its large specific surface area, low density, high electrical conductivity, and high mechanical strength, graphene has been widely used recently for energy and environmental studies. Macroscopic three-dimensional (3D) graphene structures have been obtained by interconnecting 2D graphene sheets that have been successfully synthesized via various methods. In addition to the inherent properties of 2D graphene, 3D graphene structures present large accessible surface areas, excellent mechanical strength and great flexibility because of their porous interconnected network structures [1,2,3,4]. Therefore, 3D graphene is widely studied in the gas adsorption and supercapacitor fields.

To address the environmental and energy issues caused by fossil fuel combustion, researchers have been focusing on developing renewable and clean energy sources. The utilization of H_2_, CH_4_, and other clean energy sources has become an urgent research topic [5]. Hydrogen, one of the most abundant elements in the universe, is an ideal green energy source. It emits no pollutants, and when combusted, it only produces water and heat. Moreover, the sources of hydrogen are very rich, and include solar, nuclear, and water energy. Hydrogen storage methods mainly include high-pressure hydrogen storage, low-temperature liquid hydrogen storage, and solid-state hydrogen storage. Compared with other storage options, storing hydrogen in solid materials using chemical reactions or physical adsorption is considered to be the most promising hydrogen storage method [6].

Compared with H_2_, CH_4_ has been a widely utilized resource, and could become the main fuel in the near future. However, crude CH_4_ often coexists with CO_2_ in mixtures such as natural gas, bio-gas and landfill gas, and CO_2_ is known to reduce the calorific value of gas, and corrode pipes and equipment, which leads to the decrease of its utilization efficiency. Furthermore, CO_2_ is also an important resource that is widely used in many fields such as beverage processing, agricultural product protection and supercritical extraction [7]. Therefore, it is very important to develop feasible CH_4_/CO_2_ separation technologies. Among the many methods for separating CH_4_ and CO_2_ from gas mixtures, the use of cellular materials as adsorbents could be a promising approach for the selective separation of CH_4_/CO_2_ gas mixtures using pressure swing adsorption processes [8]. Adsorption can not only be applied over a wide temperature range, but it is also a regenerable process, and regeneration requires little energy consumption. The performance of adsorbents is generally evaluated in terms of adsorption capacity, adsorption selectivity, adsorption rate, adsorbent stability, and preparation cost. To date, various solid sorbents have been developed, including carbon-based materials, zeolites, hydrotalcite-like compounds, metal oxides, and metal organic frameworks [9]. However, these conventional materials exhibit drawbacks, such as a low surface-area-to-volume ratio, and low adsorption selectivity and capacity for CO_2_.

To address the energy crisis, the first step is developing new energy. However, the key for the next step is better storage and utilization. Electrochemical energy storage systems, including lithium ion batteries and supercapacitors, are currently the most important energy storage systems. Lithium ion batteries present larger specific capacitance, faster charge-discharge rates, but poor stability [10]. Supercapacitors are a new type of high-efficiency, green energy storage devices intermediate between traditional capacitors and rechargeable batteries. Since supercapacitors exhibit large specific capacitance values, fast charge-discharge rates, long cycle lives, are non-polluting to the environment, and are extremely safe, they are advantageous compared with other energy sources [11,12]. According to their charge storage mechanisms and electrode active materials, supercapacitors can be classified into electric double layer capacitors (EDLCs) and pseudocapacitors. As a common electrode material for EDLCs, carbon-based materials present the advantages of simple process, affordability and good cycle stability. However, the specific capacitance of conventional carbon-based materials is low. Fortunately, the successful stripping of graphene in 2004 presented new opportunities for the efficient storage of supercapacitors. On the one hand, the theoretical specific capacitance of graphene is high [13,14,15]. On the other hand, it is easy to design electrode materials featuring different structures and properties using the edge structure of graphene, which is an sp^2^ hybrid carbonaceous material. Lately, graphene-based aerogels (GAs) have been gaining increasing interest due to their large specific surface areas, large pore volumes, and suitable pore size distributions and hydrophobicity [16,17,18,19].

Graphene-based aerogels are a type of 3D network structures featuring both aerogel and graphene properties, which are obtained by overlapping graphene molecules. They are mainly used in the fields of adsorption, catalysis, and energy storage, because of their excellent surface physical properties. The synthesis methods of GAs mainly include the template, chemical crosslinking and hydrothermal reduction methods [3,20]. Compared with other preparation methods, the hydrothermal reduction method relies on the self-assembly of graphene without adding any reactants. Moreover, no by-products need to be removed during the preparation process, nor does the process represent a source of pollution. This method can preserve the original properties of graphene and retain them when producing GAs. Nevertheless, graphene oxide (GO) hydrogel directly prepared by hydrothermal reduction presents a relatively low degree of reduction [21]. Large numbers of free oxygen-containing functional groups can form during hydrothermal reduction, and therefore, the thus-obtained GAs cannot fully reflect the excellent characteristics of graphene. This problem used to be solved typically by adding a binder added to GO during the preparation process. In earlier studies, synthetic polymers, such as propylene carbonate, p-phenylenediamine, and polyethyleneimine, were used as binders to crosslink GO when preparing GAs [22,23]. However, obtaining most of these synthetic polymers involved consuming fossil energy, which not only indirectly caused irreversible damage to the environment, but also contributed to the gradual depletion of the petrochemical resources, which led to increasing the scarcity of raw materials. Therefore, developing and utilizing biodegradable polymer materials instead of synthetic ones would be an effective solution. Glucose is the most widely distributed monosaccharide in nature, and it presents good biocompatibility, degradability, and crosslinkability. Therefore, in this study we prepared glucose/graphene-based aerogels (G/GAs) using the hydrothermal reduction method, employing GO and glucose as the framework and the binder, respectively. The effects of the glucose/GO mass ratio on the pore structure and adsorption performance of G/GAs were discussed, and the electrochemical properties of G/GAs were analyzed.

## 2. Experimental

### 2.1. Materials

Nano-graphite powder and glucose were purchased from Sinopharm Chemical Reagent Co., Ltd. (Shanghai, China). Potassium permanganate (KMnO_4_), glacial acetic acid (HAc), sodium hydroxide (NaOH), sodium nitrate (NaNO_3_), trimethylcarbinol, and ethanol were purchased from Tianjin Chemical Factory (Tianjin, China). All chemicals were of analytical grade (AR) and were used as received. Double-distilled water was used in all experiments.

### 2.2. Preparation of G/GAs

GO was synthesized from natural graphite using the modified Hummer’s method [24]. For a typical reaction, 40 mL of 2 mg mL^−1^ GO suspension was prepared by sonication in water for 2 h. Then, 40 mL glucose solutions of different concentrations (2, 4, 6, 8, 10, and 15 mg mL^−1^) were added to the above dispersion under sonication. The obtained solution was then transferred to a 100 mL Teflon-lined stainless steel autoclave and was hydrothermally treated at 453 K for 18 h. Then, the autoclave was naturally cooled to 298 K, and flocculent graphene/glucose hydrogel materials were obtained. The resulting samples were blotted using filter paper to remove the surface-adsorbed water. Trimethylcarbinol was added to the samples, which were again blotted using filter paper to remove the excess solution. The thus-obtained hydrogels were then freeze-dried for 24 h to completely remove the trimethylcarbinol. Finally, the unactivated G/GAs were obtained, and were labeled as H-1, H-2, H-3, H-4, H-5, and H-6. High-temperature activation of G/GAs was performed at a heating rate of 5 K min^−1^, flow rate of 166 cm^3^ min^−1^ in N_2_, and by increasing the system temperature to 1073 K while maintaining a 200 cm^3^ min^−1^ CO_2_ flow 2 h. The obtained G/GAs were labeled as HA-1, HA-2, HA-3, HA-4, HA-5, and HA-6.

### 2.3. Characterization

The N_2_ adsorption–desorption isotherms of the samples were obtained at 77 K using a specific surface porosity analyzer 3H-2000PS2 (BeiShiDe. Co., Beijing, China). The samples were degassed at 473 K for 12 h before measurements. The morphology of G/GAs was investigated using a scanning electron microscopy (SEM, S-4200/Hitachi, Tokyo, Japan) instrument. The XRD pattern and Raman spectra of the samples were obtained using an Aeris (PANalytical, Almelo, The Netherlands) instrument and a LabRAM HR800 (Horiba Jobin Yvon, Paris, France) apparatus, respectively.

### 2.4. CO_2_, CH_4_ and H_2_ adsorption measurements

Gas adsorption measurements were carried out using a 3H-2000PS2 specific surface porosity analyzer (BeiShiDe. Co., Beijing, China). The CO_2_ adsorption isotherms of the samples were obtained using the volumetric method at 298 K in the 0–1 bar pressure range. The CH_4_ and H_2_ adsorption isotherms of the samples were also obtained using the volumetric method at 298 and 77 K, respectively, in the 0–1 bar pressure range.

### 2.5. Electrochemical measurements

Cyclic voltammetry (CV), galvanostatic charge-discharge (GCD), electro-chemical impedance spectroscopy (EIS) measurements and cyclic stability analysis of HA-2 were carried out on a CHI660E electrochemical station (Shanghai Chenhua, Shanghai, China). All electrochemical measurements were performed in a three-electrode electrochemical cell setup using a platinum electrode as a counter electrode, a saturated calomel electrode as a reference electrode, and a nickel foam electrode (1 mm × 10 mm × 40 mm) coated with 1 mg mixture of HA-2, acetylene black, and poly vinylidene-fluoride binder at an 8:1:1 mass ratio as a working electrode in a 6 M KOH electrolyte solution [19]. The specific capacitance (*C_s_*), energy density (*E*), and power density (*P*) of the symmetric device were calculated using the following equations:(1)$$C_{s}=\frac{I\times \Delta t}{m\times \Delta V}$$
(2)$$E=\frac{C_{s}\times {\left(\Delta V\right)}^{2}}{2\times 3.6}$$
(3)$$P=\frac{E\times 3600}{\Delta t}$$
where *I* is the current density (A g^−1^), Δ*t* is the discharge time (s), m is the mass of active material (g), and Δ*V* is the potential window (V).

## 3. Results and Discussion

The experimental procedure for the synthesis of G/GAs is illustrated in Figure 1. The prepared GO nanosheets were uniformly mixed with glucose particles under ultrasonic dispersion. After the hydrothermal reaction and freeze drying process, the lasagna-like unactivated G/GAs were synthesized using graphene nanosheets as matrix and glucose nanoparticles as binder between the layers. The G/GAs presented a spongy hierarchical pore size architecture after CO_2_ activation. To observe the surface structure of the prepared samples, SEM images were obtained, and are presented in Figure 2. According to the various sizes of the GO nanosheets (Figure 2a), H-2 presented the relative size in Figure 2b,c. In particular, this layered structure of obtained using glucose as binder to facilitate the inter-connection of graphene sheets exhibited the fluffy formation and certain roughness on the surface of H-2 in Figure 2c. This indicated that the self-connecting cross-linked 3D structure between graphene and glucose (Figure 2d) with the high mechanical strength could provide more active sites on surface. The framework, which consisted of the matrix and binder was very dense, featured a well-proportioned architecture, and maintained the anthill-like hierarchical pores perfectly.

The integration of the macroporous and mesoporous structures of G/GAs was confirmed using N_2_ adsorption–desorption measurements. The isotherms in Figure 3a are type II and type IV isotherms, and suggest the occurrence of a multilayer adsorption process on a macroporous solid at relatively high pressures (relative pressure (*P*/*P*_0_) = 0.85–0.95). The adsorption of N_2_ could not be observed at *P*/*P*_0_ < 0.4, which implied that no micropores were present in these samples. In addition, the relatively narrow hysteresis loop between the adsorption and desorption branches (at *P*/*P*_0_ = 0.4–0.8) suggested a highly uniform pore size and no pore-blocking effects during desorption because of the narrow pores [25]. The curves for all samples were compared and revealed that GO promoted the development of mesoporosity in the samples through glucose reduction at different concentrations. Figure 3b illustrates that the pore size distribution is narrow for all the samples. The Barrett-Joyner-Halenda (BJH) method revealed a narrow distribution of mesopores centered at 2–5 nm and a broad distribution of larger pores centered at 50–90 nm. The number of mesopores approximately 2–5 nm in size decreased gradually as the glucose concentration increased. The mean pore diameter of G/GAs could be finely controlled using the glucose reduction method for preparing mesoporous samples [16].

As illustrated in Table 1, the total pore volume of the samples increased as the mesopore volume increased, which confirmed the hierarchically meso-macroporous structures of the samples. The H-2 sample exhibited superior textural parameters including high specific surface area, large pore volume, and the largest mesopore volume formed by hydrothermal reduction compared to the other analyzed samples. The adsorption performance of the unactivated H-1 G/GAs was higher than that of the H-2 sample because H-1 presented a narrow size distribution of mesopores centered at 3 nm (Figure 3b) and almost no macropores (Table 1). Therefore, the specific surface areas of all samples in this study mainly consisted of narrow mesopores. Table 2 illustrates that at the high activation temperature of 1073 K, the specific surface area, pore volume, and adsorption capacity increased for all the samples, compared to the unactivated samples, demonstrating that the CO_2_ activation method was extremely efficient for improving the surface adsorption properties of G/GAs. We determined that G/GAs presented high specific surface areas of up to 763 m^2^ g^−1^, good adsorption capacity for CO_2_ of 76.5 mg g^−1^ at 298 K, and the highest adsorption capacities of 16.8 mg g^−1^ for CH_4_ at 298 K and 12.1 mg g^−1^ for H_2_ at 77 K.

Raman spectra (Figure 4) presented the typical D-band (1328 cm^−1^) and G-band (1599 cm^−1^), which were attributed to the vibrations of disordered carbon atoms presenting defects and sp^2^ hybrid carbon atoms in the 2D hexagonal lattice, respectively. The *I*_D_/*I*_G_ ratios assigned to GO (1.13), H-2 (1.17), and HA-2 (1.13) indicated the high degree of graphitization of GO and its fabricated materials, while demonstrating that G/GAs maintained the intrinsic properties of GO without severe fluctuation. The absence of the 2D-band from the spectra of H-2 and HA-2 indicated the amorphous nature of G/GAs, whereas the appearance of the 2D-band in the spectrum of GO indicated its intrinsic crystallinity [26]. The XRD patterns illustrating the crystalline structures and phases of H-2 and HA-2 are depicted in Figure 5. The prepared materials have been demonstrated to present identical crystalline structure, exhibiting peaks corresponding to the (002) and (422) planes which were attributed to the graphene of the graphite planes, at 2θ of 25.48° and 44.15°, respectively. The intensities of the G/GAs peaks in the Raman spectra and XRD patterns were higher after CO_2_ activation, which indicated that the CO_2_ activation method did not only enhance the specific surface and porosity, but also increased the degree of graphitization of G/GAs [27,28].

Based on the above results, we proposed a method for manufacturing G/GAs. First, the heat treatment in water constituted the origin of the formation of mesopores (2–5 nm) and macropores (>50 nm). The GO sheets were rich in oxygen-containing groups, such as hydroxyl groups. Upon heating, some of the oxygen-containing groups could be converted into G/GAs, which could escape from the GO sheets and create nanopores between the sheets. Second, glucose as the reducing agent underwent carbonization at high temperatures and reacted with GO to form mesoporous structures [29].

The adsorption capacities were reported as excess uptake in the CO_2_ and CH_4_ adsorption isotherms of all samples at two different temperatures (298 and 77 K) under the pressure of 1 bar pressure (Figure 6 and Figure 7, respectively). Recent research has indicated that a large specific surface area and large total pore volume are not decisive factors for high CO_2_ adsorption capacity, and that the pore size distribution of samples, which was centered at 3 nm, was more effective than others for enhancing CO_2_ adsorption [22,30]. As can be seen from Figure 6 and Table 2, HA-4 exhibited the highest CO_2_ adsorption capacity of 76.5 mg g^−1^ at 298 K and 1 bar. However, the CO_2_ adsorption capacity of HA-2 was very close to it (73.0 mg g^−1^), thus indicating the superiority of the material obtained when 4 mg mL^−1^ glucose was used in the fabrication process. Masika et al. have synthesized carbon aerogels using resin and metal salt as matrix and template, respectively and obtained materials featuring for CO_2_ adsorption capacities in the 44.0–96.8 mg g^−1^ range [22]. The adsorption capacity of G/GAs for CO_2_ was within that range, therefore demonstrating that the adsorption properties of G/GAs were typical for carbon-based aerogels.

Sample HA-2 presented the highest CH_4_ adsorption capacity of 16.8 mg g^−1^ at 298 K and 1 bar (Figure 6). Andrea et al. synthesized carbon aerogels featuring CH_4_ adsorption capacities in the 9.4–23.9 mg g^−1^ range, which indicated that G/GAs were good adsorbents for CH_4_ [31]. However, the adsorption capacity of HA-2 was lower than that of HA-4 at pressures smaller than 0.9 bar, mainly due to the narrow mesoporous structure of the HA-2 sample. Moreover, based on the CO_2_ and CH_4_ adsorption isotherms, these adsorbents could also be practical for separating CO_2_/CH_4_ gas mixtures. As can be seen from Figure 6 and Figure 7, the adsorption capacity of G/GAs for CO_2_ was significantly higher than that for CH_4_. This occurred because G/GAs contain more micropores, and feature a microporous linear structure. From a molecular polarity perspective, CO_2_ is a type of non-polar molecule. However, its molecular structure is a quadrupole one featuring a certain degree of polarity, therefore CO_2_ is sensitive to the presence of polar groups on solid surfaces. The CH_4_ molecule is a type of non-polar molecule presenting spherical symmetry, and its molecular dynamic diameter is larger than that of CO_2_. According to the principle of preferential adsorption of small pore size molecules, the amount of adsorbed CO_2_ would be larger than that of CH_4_ [32,33].

The H_2_ adsorption isotherms of all samples (Figure 8) indicated that the equilibrium amounts of adsorbed H_2_ increased as the pressure of the system increased. As the adsorption sites approached saturation, the slopes of the isotherms decreased at higher pressures. The HA-2 sample also presented the highest H_2_ uptake of all analyzed samples: 12.1 mg g^−1^ at 77 K and 1 bar (Table 2). Although the H_2_ uptake value of HA-2 was much lower than the U.S. Department of Energy (DOE) target of 65 mg g^−1^ (at present, no experimental H_2_ storage materials meet the requirements for practical applications set by the U.S. DOE), HA-2 exhibited ultrahigh H_2_ adsorption efficiency, as indicated by the uptake per unit specific surface area of 1.14 × 10^−3^ wt % m^2^ g^−1^, which was close to the upper limit of H_2_ adsorbed in a monolayer [34].

The electrochemical properties of HA-2 at different scanning rates were analyzed utilizing via a three-electrode test system using 6 M KOH as an electrolyte. We were able to demonstrate that the CV curves of HA-2 at different scanning rates (Figure 9) did not present any significant redox peaks, and presented the standard features of the typical graphene structure used as an electrode material. As the scan rate was lower than 200 mV s^−1^, the rectangular characteristic of the electric double layer energy storage could be maintained, and was much higher than that of conventional graphene-based materials. This indicated that the 3D prepared materials featured good stability, and adding the binder was necessary [35]. The area of the CV curves increased as the scan rate increased, which suggested that G/GAs presented excellent rate capability. Typically, the nanoporous architecture of porous EDLCs electrodes presents short charge transportation distances, thus providing excellent charge transport paths during the charge-discharge process. The electrical double layer can form fast, and it can be quickly and effectively restructured at switching potentials. Moreover, the electrical double layer could subsequently reach a steady state which would result in rectangular-shaped CV curves. This indicated that the charge storage mechanisms of G/GAs resulted from the reversible adsorption of the electrolyte ions onto the surface of carbon at the electrolyte/electrode interface. The rectangular like shapes of the present CV curves of G/GAs indicated fast ion transportation, and high ion diffusion rate within the electrode structure and at the electrode/electrolyte interface at various potentials [36,37]. In addition, the CV curves also maintained their rectangular shape at different scan rates, thus demonstrating the good stability of the charge storing process.

The linear shape of the GCD curves at different scan rates (Figure 10) also confirmed the properties of HA-2 as EDLCs. When the voltage was gradually reduced to −1.0 V, the shapes of the GCD curves became concave, indicating high graphitization, which was confirmed by both the XRD patterns and Raman spectra [38,39]. The maximum specific capacitance of 305.5 F g^−1^ was obtained at 1 A g^−1^. In particular, it was observed that the discharging section of the curve was asymmetric compared to the charging one since the discharging time was longer than the charging one. This could be attributed to self-discharging, which usually occurs by redistributing the charges on a porous electrode material at low current densities since the charge storing process in porous carbon electrodes occurs more rapidly on the outer surface of the electrode than in the bulk of the electrode.

Electrochemical impedance spectroscopy (EIS) is generally utilized for supercapacitor applications as an important electrochemical characterization, because it evaluates the impedance behavior including the resistive and capacitive behaviors, particularly at electrode/electrolyte interfaces [40]. The AC impedance spectrum in Figure 11 demonstrates the reasonably good impedance of HA-2 in the 0.01 Hz to 100 kHz frequency range. To identify the charge transfer behavior of materials using Nyquist plots, an equivalent-circuit diagram was generated for the EIS plot, which helped determine that the related resistance of HA-2 was 12.4 Ω. In the high frequency region, the Nyquist plot of HA-2 presented a well-fitted semicircle, while the low frequency region exhibited good linearity. The line is close to 45°, which fits the standard of the mixed control between charge transfer and diffusion processes in electrochemical system. To sum up, G/GAs presents good electrochemical properties and stability and can be used as electrode materials for EDLCs [41,42].

To evaluate the stability of G/GAs electrodes, the specific capacitance retention (Figure 12) was repeatedly measured by GCD for 10,000 cycles at 10 A g^−1^. It was observed that HA-2 exhibited a capacitance retention of 98.5%, Miao et al. synthesized a magnetic N-doped carbon aerogel featuring a capacity of 185.3 F g^−1^ and 90.2% capacitance retention after 5000 cycles at 10 A g^−1^, which indicated that the prepared G/GAs presented very high cyclic stability, therefore demonstrating that the porous structure of G/GAs was an important factor for efficient ion diffusion [43]. The EDLCs device performances of G/GAs were examined using two HA-2 based electrode as positive and negative electrode for practical application, the results showed the specific capacitance of 135.2 F g^−1^, good cycle stability, and increased wettability after 10,000 cycles at 10 A g^−1^. In addition, the energy and power densities are also important properties of electrode materials. As displayed in the Ragone plots in Figure 13, the energy density of HA-2 linearly decreases from 42.43 to 2.11 Wh kg^−1^ for the HA-2 based supercapacitor as the power density increases from 634 to 6333 W kg^−1^. These values were much higher than those of graphene-based hybrid electrodes [44], and were very close to those of two other hybrid electrodes reported in the literature [45,46]. However, these other electrodes did not contain added metal oxides, which illustrated that G/GAs could be used in a wider range of practical applications as an electrochemical material. To ensure the EDLCs device performances of the G/GAs based electrode, a symmetrical electrochemical capacitor device based on two G/GAs based electrode, as positive and negative electrodes, was measured in 6 M KOH. The results were showed in Appendix A.

## 4. Conclusions

Herein, a facile synthesis of glucose/graphene-based G/GAs was obtained showing remarkable gas adsorption capacity and electrochemical performance of EDLCs with higher capacity and better high-rate performance. In this configuration, more micropores with polar groups on the G/GAs surface feature the relative sensitivity for the non-polar molecule CO_2_, and graphene layers connected by glucose-based spheres with nanopores centered at 3 and 50–90 nm work together to render remarkable absorbability and a high transport rate for electrons. This indicated the potential of G/GAs for applications in removal of CO_2_ from gas mixtures (such as natural gas, bio-gas, and landfill gas, etc.) by reason of the adsorption selectivity. Further studies toward the development of GAs featuring higher specific surface areas and more promising high electrochemical properties are worth conducting for active sites and high conductivity in the catalyst carrier field.

## Figures and Tables

**Figure 1 polymers-11-00040-f001:**
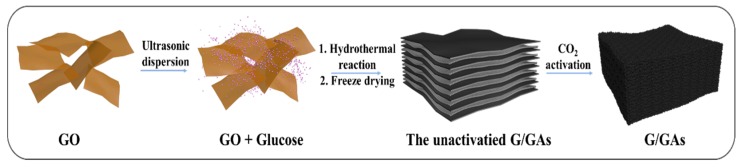
Schematic illustration for preparation of G/GAs.

**Figure 2 polymers-11-00040-f002:**
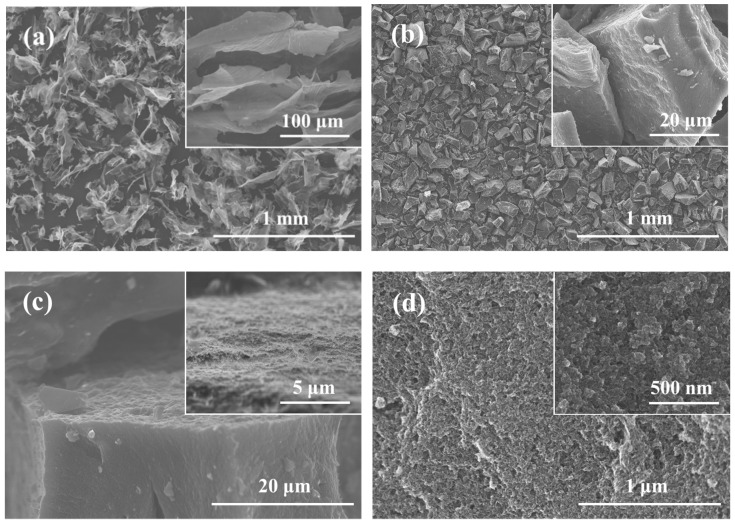
SEM images of GO (**a**), H-2 (**b**–**c**: 4 mg mL^−1^ glucose), HA-2 (**d**: 4 mg mL^−1^ glucose & CO_2_ activation at 1073 K).

**Figure 3 polymers-11-00040-f003:**
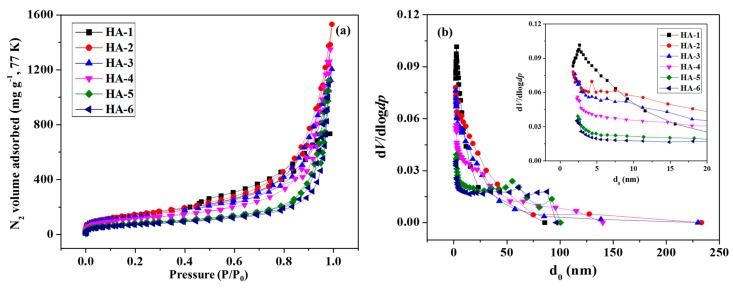
N_2_/77 K full isotherms (a) and pore size distributions (b) of G/GAs.

**Figure 4 polymers-11-00040-f004:**
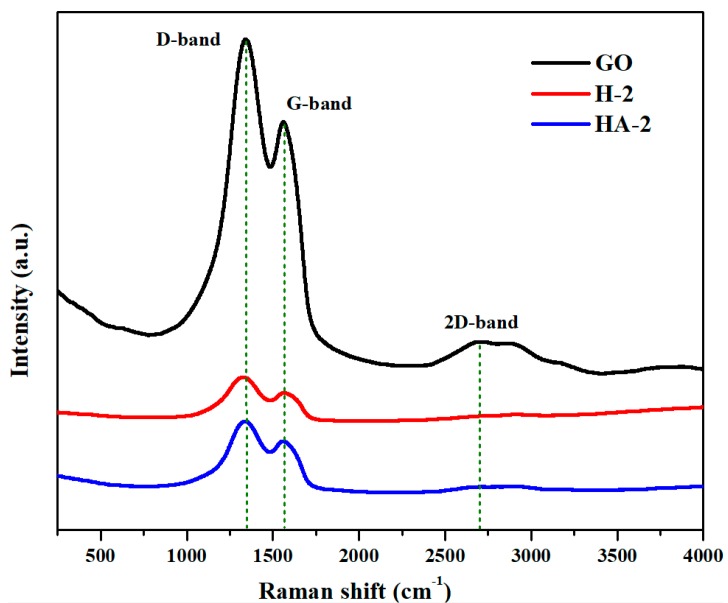
Raman spectra of GO, H-2 (4 mg mL^−1^ glucose) and HA-2 (4 mg mL^−1^ glucose & 1073 K CO_2_ activation).

**Figure 5 polymers-11-00040-f005:**
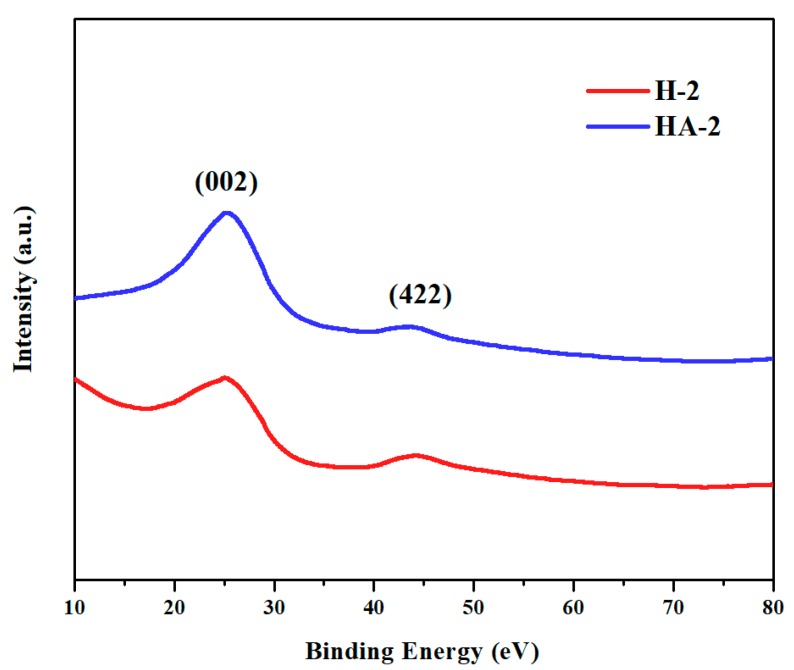
XRD patterns of H-2 (4 mg mL^−1^ glucose) and HA-2 (4 mg mL^−1^ glucose & 1073 K CO_2_ activation).

**Figure 6 polymers-11-00040-f006:**
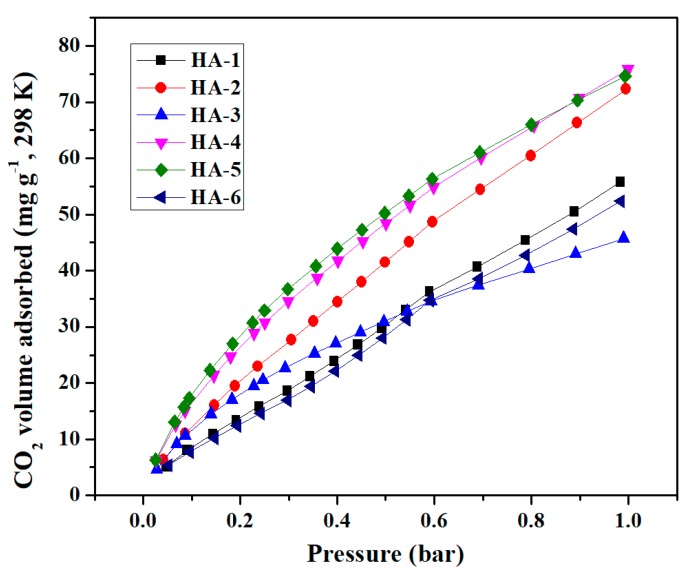
CO_2_/298 K adsorption isotherms of G/GAs.

**Figure 7 polymers-11-00040-f007:**
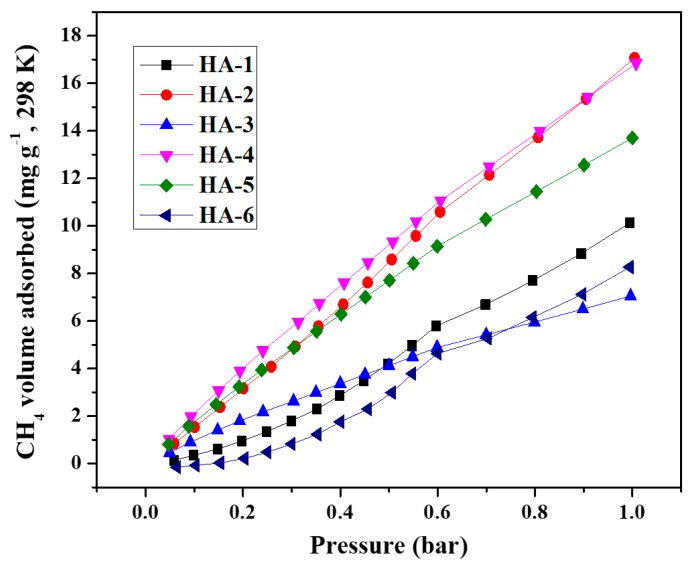
CH_4_/77 K adsorption isotherms of G/GAs.

**Figure 8 polymers-11-00040-f008:**
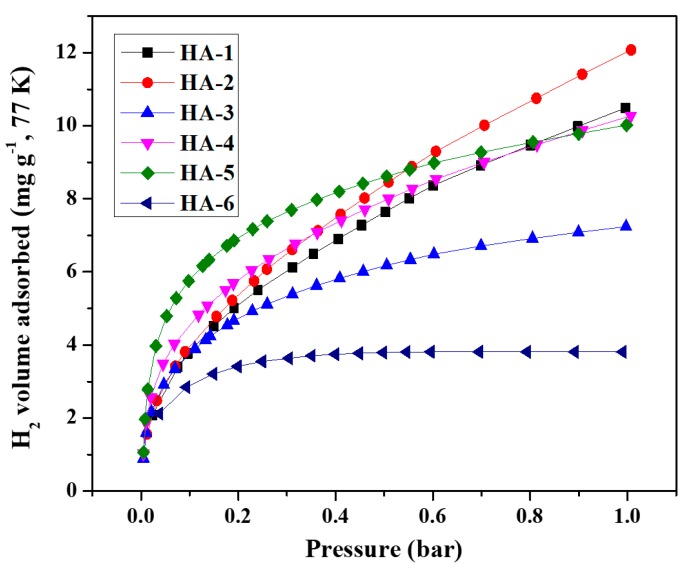
H_2_/77 K adsorption isotherms of G/GAs.

**Figure 9 polymers-11-00040-f009:**
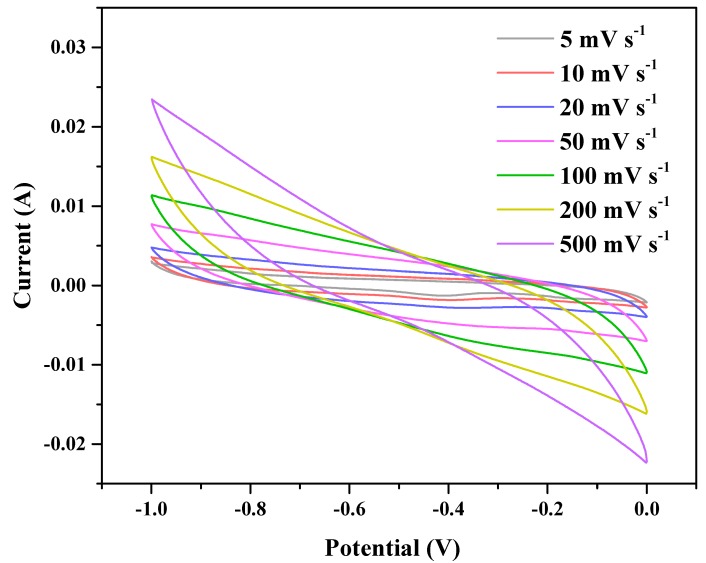
CV curves of HA-2 (4 mg mL^−1^ glucose & 1073 K CO_2_ activation) at different scan rates.

**Figure 10 polymers-11-00040-f010:**
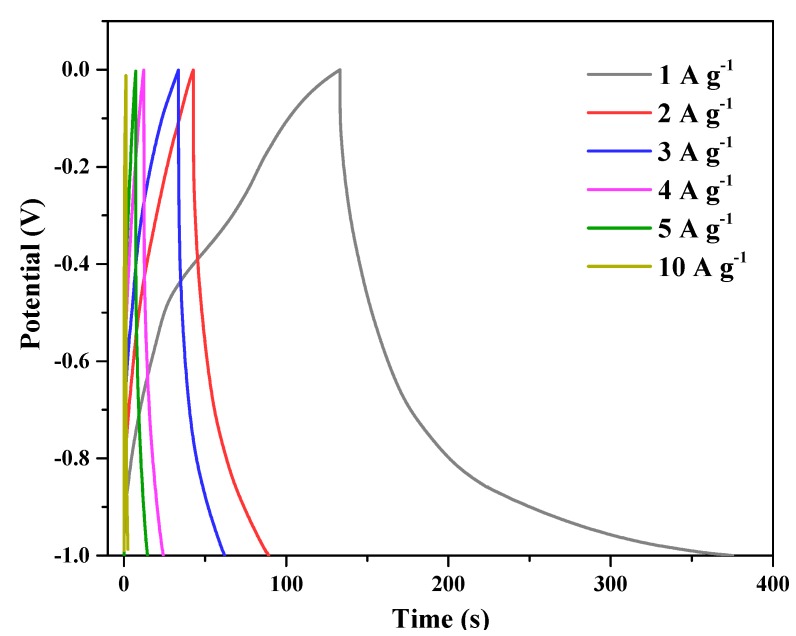
Galvanostatic charge-discharge tests of HA-2 (4 mg mL^−1^ glucose & 1073 K CO_2_ activation) at different current densities.

**Figure 11 polymers-11-00040-f011:**
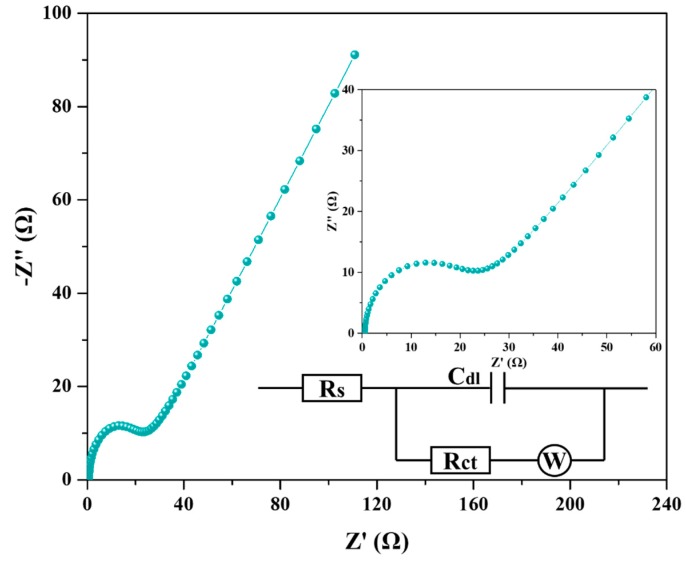
Nyquist plots of HA-2 (4 mg mL^−1^ glucose & 1073 K CO_2_ activation) from 0.01 Hz to 100 kHz, inset shows the magnified high-frequency regions.

**Figure 12 polymers-11-00040-f012:**
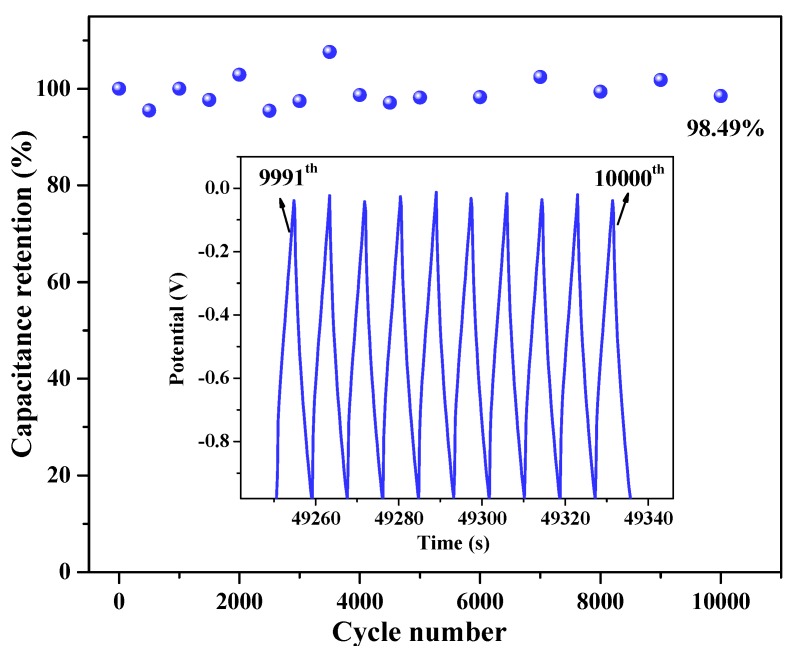
Cyclic stability of HA-2 (4 mg mL^−1^ glucose & 1073 K CO_2_ activation).

**Figure 13 polymers-11-00040-f013:**
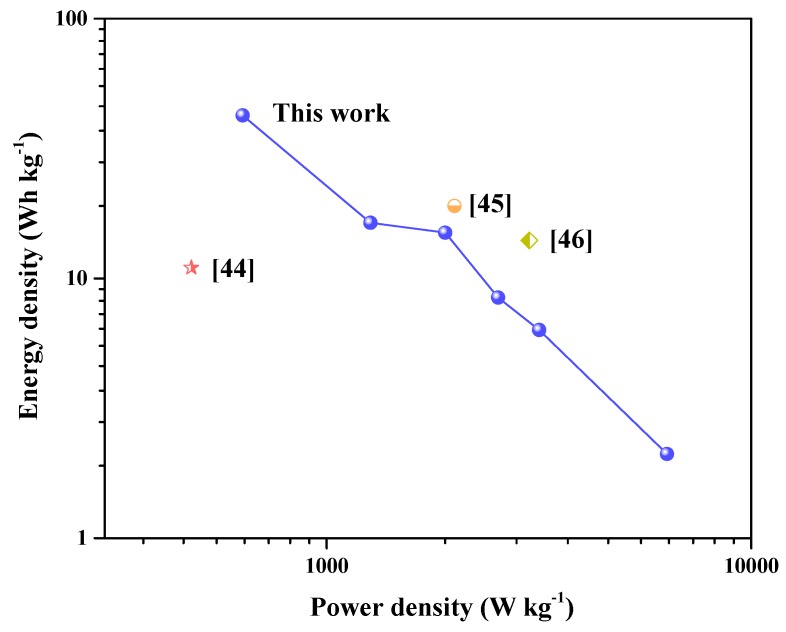
Ragone plots of HA-2 (4 mg mL^−1^ glucose & 1073 K CO_2_ activation). Inset points are corresponding values in the literature.

**Table 1 polymers-11-00040-t001:** Pore structure and gas adsorption parameters of the unactivated G/GAs.

Samples	S_BET_ ^a^ (m^2^ g^−1^)	V_Total_ ^b^ (cm^3^ g^−1^)	V_Meso_ ^c^ (cm^3^ g^−1^)	D_BET_ ^d^ (nm)	CO_2 Uptake_ ^e^ (mg g^−1^)	CH_4 Uptake_ ^f^ (mg g^−1^)	H_2 Uptake_ ^g^ (mg g^−1^)
H-1	519	1.14	1.29	8.76	36.34	5.17	0.59
H-2	522	2.37	2.53	18.18	32.12	5.07	0.53
H-3	476	1.86	2.01	15.63	29.83	3.43	0.42
H-4	412	2.09	2.20	20.23	23.50	2.26	0.36
H-5	292	1.74	1.82	23.75	19.02	2.10	0.30
H-6	256	1.69	1.76	26.32	17.82	2.14	0.21

^a^ Specific surface area calculated by Brunauer-Teller-Emmett (BET) method (Error value: ±2.63%). ^b^ Total pore volume at relative pressure of *P*/*P*_0_ = 0.99. ^c^ Mesopore volume calculated from N_2_ sorption isotherms by BJH method. ^d^ Average of pore size distribution by BET method. ^e^ CO_2_ uptake capacity at 298 K and 1 bar. ^f^ CH_4_ uptake capacity at 298 K and 1 bar. ^g^ H_2_ uptake capacity at 77 K and 1 bar.

**Table 2 polymers-11-00040-t002:** Pore structure and G/GAs adsorption parameters of G/GAs.

Samples	S_BET_ ^a^ (m^2^ g^−1^)	V_Total_ ^b^ (cm^3^ g^−1^)	V_Micro_ ^c^ (cm^3^ g^−1^)	V_Meso_ ^d^ (cm^3^ g^−1^)	D_BET_ ^e^ (nm)	CO_2 Uptake_ ^f^ (mg g^−1^)	CH_4 Uptake_ ^g^ (mg g^−1^)	H_2 Uptake_ ^h^ (mg g^−1^)
HA-1	749	1.74	0	1.67	9.28	56.4	10.0	10.5
HA-2	763	3.06	0	3.22	16.04	73.0	16.8	12.1
HA-3	632	2.23	0	2.34	14.10	46.2	7.1	7.2
HA-4	560	1.79	0.04	1.84	12.78	76.5	16.6	10.2
HA-5	522	1.59	0.06	1.60	12.17	75.3	13.5	10.0
HA-6	544	1.21	0.14	1.13	8.89	52.9	8.2	3.7

^a^ Specific surface area calculated by BET method (Error value: ±2.63%). ^b^ Total pore volume at relative pressure of *P*/*P*_0_ =0.99. ^c^ Micropore volume calculated from N_2_ sorption isotherms by T-Plot method. ^d^ Mesopore volume calculated from N_2_ sorption isotherms by BJH method. ^e^ Average of pore size distribution by BET method. ^f^ CO_2_ uptake capacity at 298 K and 1 bar. ^g^ CH_4_ uptake capacity at 298 K and 1 bar. ^h^ H_2_ uptake capacity at 77 K and 1 bar.

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
