# Peer review of "Glucose/Graphene-Based Aerogels for Gas Adsorption and Electric Double Layer Capacitors"

_polymers, 2018, doi:10.3390/polym11010040_

Round 1
Reviewer 1 Report
Summary: In this work, graphene-based aerogels were prepared by gelation using glucose as a binder followed by hydrothermal reduction and Co2 activation. The obtained material was tested regarding CO2, CH4 and H2 adsorption capacity as well as its electrochemical performance for its applications as gas adsorbent and/or supercapacitor.
Overall opinion: The paper is written in correct English, the research methodology is interesting and the quality of the manuscript is high. The novelty of the approach is the use of glucose as a sustainable binder for the processing of graphene aerogels. Some aspects in the Discussion section need to be clarified. Minor revisions are needed before being accepted in Polymers journal.
Specific comments:
- Introduction:
+There is a mistake in the whole section where the word "gas" is replaced by "GAs" or "G/GAs" terms. Please modify it.
+Line 29: please specify the other methods used for the synthesis of 3D-graphene structures.
+Line 56: ref [9] does not match with the content of this sentence, please revise the citation.
+Lines 73: please replace "has" by "have"
- Experimental:
+Section 2.1: the reagents in this section do not match with the materials used for the processing of the aerogels (e.g., glucose or trimethylcarbinol are missing). Please revise it.
+Lines 112-113: please specify the shape and appearance of the obtained gels
+L122-123: please indicate the degassing conditions for the BET analysis.
+L128-131: the supplier of the supercritical equipment or elements should be specified.
-Results and discussion:
+Table 1: please provide the error values.
+Figure 2: the SEM image do not have enough magnification to see the mesopores. Please remove it.
+Lines 213-218: the proposed mechanism of the hydrothermal treatment is mere speculation with the results provided in the manuscript. Please specify whether this treatment is innovative or already used before and provide further evidences to support your hypothesis.
+Lines 219-226: please compare the CO2 and CH4 adsorption values with other adsorbents (e.g., other aerogels) in the literature.
+Line 237: it is stated that G/GAs aerogels have micropores whereas in line 148 it is mentioned that there are no micropores. Please clarify this.
+Line 248: please specify the DOE target value.
+Lines 285-299: please compare the electrochemical values with other in the literature, e.g., with other aerogels like in [Carbohydrate Polymers 189 (2018) 304-312]
Author Response
Response to Reviewer 1 Comments
Point 1: There is a mistake in the whole section where the word "gas" is replaced by "GAs" or "G/GAs" terms. Please modify it.
Response 1: Thanks for the referee’s kind advice. Those mistakes have already corrected in line 33、45.
Therefore, 3D graphene is widely studied in the fields of gas adsorption and supercapacitor.
However, the crude CH4 is often coexisted with CO2 in mixtures such as natural gas, bio-gas and landfill gas. Since CO2 makes the calorific value of gas reduced, and corrosion of pipes and equipment caused, which leads to decrease of the utilization efficiency.
Point 2: Line 29: please specify the other methods used for the synthesis of 3D-graphene structures.
Response 2: Thanks for the referee’s kind advice. The methods of the synthesis of 3D-graphene structures were specified in line 78-79.
The synthesis methods of GAs mainly include template, chemical crosslinking and hydrothermal reduction method
Point 3: Line 56: ref [9] does not match with the content of this sentence, please revise the citation.
Response 3: Thanks for the referee’s good evaluation and kind suggestion. The citation of Ref.[9] was revised.
Chao, C.; Wha-Seung, A. CO2 adsorption on LTA zeolites: Effect of mesoporosity. Appl. Surf. Sci. 2014, 311, 107-109.
Point 4: Lines 73: please replace "has" by "have".
Response 4: Thanks for the referee’s good evaluation and kind suggestion. The word “has” was replaced by “have” in line 73.
Graphene aerogels (GAs) have lately gained increasing interest due to their large specific surface area, large pore volume, and suitable pore size distribution and hydrophobicity
Point 5: Section 2.1: the reagents in this section do not match with the materials used for the processing of the aerogels (e.g., glucose or trimethylcarbinol are missing). Please revise it.
Response 5: Thanks for the referee’s good evaluation and kind suggestion. The mistakes were corrected in line 101、103.
Nano-graphite powder and glucose were purchased from Sinopharm Chemical Reagent Co., Ltd. Potassium permanganate (K2MnO4), glacial acetic acid (HAc), sodium hydroxide (NaOH), sodium nitrate (NaNO3), trimethylcarbinol and ethanol were purchased from Tianjin Chemical Factory (China).
Point 6: Lines 112-113: please specify the shape and appearance of the obtained gels.
Response 6: Thanks for the referee’s good evaluation and kind suggestion. The shape of the obtained gels was specified in line 113.
The obtained solution was then transferred to a 100 mL Teflon-lined stainless steel autoclave and was hydrothermally treated at 180℃ for 18 h. Then, the autoclave was naturally cooled to room temperature, and the flocculent graphene/glucose hydrogel materials were obtained.
Point 7: L122-123: please indicate the degassing conditions for the BET analysis.
Response 7: Thanks for the referee’s good evaluation and kind suggestion. The degassing conditions for the BET analysis were indicated in line 123-124.
The N2 adsorption-desorption isotherms of the samples were obtained at 77 K using specific surface porosity analyzer 3H-2000PS2 (BeiShiDe. Co., China). The samples were degassed at 473 K for 12 h before the measurement.
Point 8: L128-131: the supplier of the supercritical equipment or elements should be specified.
Response 8: Thanks for the referee’s good evaluation and kind suggestion. The supplier of the gas adsorption was specified in line 128-129.
The gas adsorption measurements were carried out by specific surface porosity analyzer 3H-2000PS2 (BeiShiDe. Co., China).
Point 9: Table 1: please provide the error values.
Response 9: Thanks for the referee’s good evaluation and kind suggestion. The company who providing BET data just gave us the error values of specific surface area. The error values were provided in the notes of Table 1 and Table 2.
a Specific surface area calculated by Brunauer-Teller-Emmett (BET) method (Error value: ±2.63%).
Point 10: Figure 2: the SEM image do not have enough magnification to see the mesopores. Please remove it.
Response 10: Thanks for the referee’s good evaluation and kind suggestion. The SEM images were improved, and the discussion was adjusted in line 161.
To observe the surface structure of the prepared sample, SEM images were obtained in Figure 2. According to various size of GO nanosheets (Figure 2a), H-2 showed the relative size in Figure 2b, 2c. In particular, this structure of layers upon layers overlay using glucose as binder to facilitate inter-connection of graphene sheets exhibited the surface of H-2 with fluffy formation and certain roughness in Figure 2c. It indicates that the 3D structure of self-connecting between graphene and glucose (Figure 2d) with the cross-linked structure effectively provides mechanical strength which would support various access to utilize in surface reaction. The framework made of matrix and binder was very dense with well-proportioned architecture, it maintained perfectly anthill-liked hierarchical pores.
Figure 2. SEM images of GO (a), H-2 (4 mg mL-1 glucose, b-c), HA-2 (4 mg mL-1 glucose & CO2 activation at 800℃, d).
Point 11: Lines 213-218: the proposed mechanism of the hydrothermal treatment is mere speculation with the results provided in the manuscript. Please specify whether this treatment is innovative or already used before and provide further evidences to support your hypothesis.
Response 11: Thanks for the referee’s good evaluation and kind suggestion. The evidence to support the hypothesis was cited in line 423.
Andrea, D.; Balázs, N.; Laura, P.N.; Dávid, S.; János, M.; Krisztina, L. Pressure resistance of copper benzene-1,3,5-tricarboxylate – carbon aerogel composites. Appl. Surf. Sci. 2018, 434, 1300-1310.
Point 12: Lines 219-226: please compare the CO2 and CH4 adsorption values with other adsorbents (e.g., other aerogels) in the literature.
Response 12: Thanks for the referee’s good evaluation and kind suggestion. The comparisons were added in line 247-249 and 255-256.
Masika, et al. [22] have synthesized the carbon aerogels via resin as matrix and metal salt as template for CO2 adsorption with capacity of 44.0-96.8 mg g-1, which covered G/GAs in that range, proving that G/GAs were typical for carbon-based aerogels.
Andrea, et al. [31] synthesized carbon aerogels of CH4 adsorption capacity with range of 9.4-23.9 mg g−1, indicating G/GAs were good adsorbent for CH4.
Point 13: Line 237: it is stated that G/GAs aerogels have micropores whereas in line 148 it is mentioned that there are no micropores. Please clarify this.
Response 13: Thanks for the referee’s kind advice. The samples of the study do exist non-micropores, but HA-4, HA-5, HA-6 have the micropores in Table 2.
Samples | SBET a (m2 g−1) | VTotalb (cm3 g−1) | VMicroc (cm3 g−1) | VMesod (cm3 g−1) | DBETe (nm) | CO2 uptakef (mg g−1) | CH4 uptakeg (mg g−1) | H2 uptakeh (mg g−1) | |||||||||
HA-4 | 560 | 1.79 | 0.04 | 1.84 | 12.78 | 76.5 | 16.6 | 10.2 | |||||||||
HA-5 | 522 | 1.59 | 0.06 | 1.60 | 12.17 | 75.3 | 13.5 | 10.0 | |||||||||
HA-6 | 544 | 1.21 | 0.14 | 1.13 | 8.89 | 52.9 | 8.2 | 3.7 | |||||||||
Point 14: Line 248: please specify the DOE target value.
Response 14: Thanks for the referee’s good evaluation and kind suggestion. The DOE target value of H2 adsorption was added in line 272.
Point 15: Lines 285-299: please compare the electrochemical values with other in the literature, e.g., with other aerogels like in [Carbohydrate Polymers 189 (2018) 304-312]
Response 15: Thanks for the referee’s good evaluation and kind suggestion. The comparison has already added in line 321-323.
It was observed that HA-2 exhibited 98.5% capacitance retention, Miao, et al. [43] synthesized magnetic N-doped carbon aerogel with the capacity of 185.3 F g−1 and 90.2% capacitance retention after 5000 cycles at 10 A g−1, indicating that the prepared G/GAs has a very high cyclic stability, meanwhile which proved the porous structure of G/GAs is an important factor for efficient ion diffusion.

Reviewer 2 Report
In this paper, authors synthesized three-dimensional glucose/graphene-based aerogels (G/GAs) via hydrothermal reduction and CO2 activation for gas adsorption and supercapacitor. This work is very interesting. The reviewer would like to recommend its publication if the authors could address the following issues.
1. Authors just described the materials for building supercapacitor electrode, but how to build the electrode?
2. Now two-electrode systems are usually needed to examine the performance of electrode materials for supercapacitors.
3. Authors claimed that SEM images indicate the 3D structure self-connecting between graphene and glucose. However from SEM images in Fig.2, it is difficult to observe the 3-dimensional structure of the sample of HA-2. Therefore, how to characterize the 3D structure of the sample?
4. When explaining the formation process of G/GAs, if authors could provide a schematic diagram, it would be better.
5. Ragone plot which shows the relationship between energy density and power density is very useful for describing the electrochemical performance of the electrode materials.
Author Response
Response to Reviewer 2 Comments
Please see the attachment
Point 1: Authors just described the materials for building supercapacitor electrode, but how to build the electrode?
Response 1: Thanks for the referee’s good evaluation and kind suggestion. The preparation of working electrode was improved in line 138, 139.
All electrochemical measurements with using a platinum electrode as counter electrode and a saturated calomel electrode as reference electrode in 6 M KOH electrolyte solution checked working electrode, made of nickl foam (1 mm × 10 mm × 40 mm) coated with 1 mg material mixing by HA-2, acetylene black and poly vinylidene-fluoride binder at 8:1:1 in a three-electrode electrochemical cell setup [19].
Point 2: Now two-electrode systems are usually needed to examine the performance of electrode materials for supercapacitors.
Response 2: Thanks for the referee’s kind advice. Yes, you are right, now two-electrode systems really are usually needed to examine the performance of electrode materials for supercapacitors. We also realized that two-electrode system is an effective method to characterize the electrochemical performance, but three-electrode system also can do what two-electrode system does, and we have no lab condition (including button battery packaging machine, vacuum glove box, and two-electrode test system, et al.) for two-electrode system temporarily. It is really sorry we can't finish the request, but we have added some data, hoping to try to improve this paper more.
Point 3: Authors claimed that SEM images indicate the 3D structure self-connecting between graphene and glucose. However from SEM images in Fig.2, it is difficult to observe the 3-dimensional structure of the sample of HA-2. Therefore, how to characterize the 3D structure of the sample?
Response 3: Thanks for the referee’s good evaluation and kind suggestion. SEM images were updated in Figure 2.
To observe the surface structure of the prepared sample, SEM images were obtained in Figure 2. According to various size of GO nanosheets (Figure 2a), H-2 showed the relative size in Figure 2b, 2c. In particular, this structure of layers upon layers overlay using glucose as binder to facilitate inter-connection of graphene sheets exhibited the surface of H-2 with fluffy formation and certain roughness in Figure 2c. It indicates that the 3D structure of self-connecting between graphene and glucose (Figure 2d) with the cross-linked structure effectively provides mechanical strength which would support various access to utilize in surface reaction. The framework made of matrix and binder was very dense with well-proportioned architecture, it maintained perfectly anthill-liked hierarchical pores.
Figure 2. SEM images of GO (a), H-2 (4 mg mL-1 glucose, b-c), HA-2 (4 mg mL-1 glucose & CO2 activation at 800℃, d).
Point 4: When explaining the formation process of G/GAs, if authors could provide a schematic diagram, it would be better.
Response 4: Thanks for the referee’s good evaluation and kind suggestion. The schematic diagram was provided in Figure 1.
Figure 1. Schematic illustration for preparation of G/GAs.
The experimental procedure of G/GAs synthesis mechanism is showed in Figure 1. The prepared GO nanosheets mixed uniformly with glucose particles under the ultrasonic dispersion. After hydrothermal reaction and freeze drying, the lasagna-like unactivated G/GAs were synthesized by graphene nanosheets as matrix with glucose nanoparticles as binder between layers and layers. G/GAs were full of spongy hierarchical pore size architecture after CO2 activation.
Point 5: Ragone plot which shows the relationship between energy density and power density is very useful for describing the electrochemical performance of the electrode materials.
Response 5: Thanks for the referee’s good evaluation and kind suggestion. The Ragone plots were provided in Figure 13.
In addition, the energy density and power density are also the important characterizations for electrode materials, displayed the Ragone plots as shown in Figure 13, the energy density of HA-2 descends linearly from 42.43 to 2.11 Wh kg-1 for HA-2-based supercapacitor as the power density increases from 634 to 6333 W kg-1, much higher than the values of graphene-based hybrid electrode [44], and very close to another two hybrid electrode [45,46], but without adding metal oxide, which illustrated that G/GAs as electrode material have a wide prospect for practical application.
Figure 13. Ragone plots of HA-2 (4 mg mL-1 glucose & 800℃ CO2 activation). Inset points are corresponding values in literatures.

Round 2
Reviewer 2 Report
Point 2: Now two-electrode systems are usually needed to examine the performance of electrode materials for supercapacitors.
Response 2: Thanks for the referee’s kind advice. Yes, you are right, now two-electrode systems really are usually needed to examine the performance of electrode materials for supercapacitors. We also realized that two-electrode system is an effective method to characterize the electrochemical performance, but three-electrode system also can do what two-electrode system does, and we have no lab condition (including button battery packaging machine, vacuum glove box, and two-electrode test system, et al.) for two-electrode system temporarily. It is really sorry we can't finish the request, but we have added some data, hoping to try to improve this paper more.
According to the manuscript (2.5. Electrochemical measurements), the authors’ lab condition is enough for the two-electrode system examination in KOH electrolyte. There are many papers describing how to do the two-electrode examination with the authors’ lab condition.
Author Response
Response to Reviewer 2 Comments
Point 1: According to the manuscript (2.5. Electrochemical measurements), the authors’ lab condition is enough for the two-electrode system examination in KOH electrolyte. There are many papers describing how to do the two-electrode examination with the authors’ lab condition.
Response 1: Thanks for the referee’s good evaluation and kind suggestion. The two-electrode system examination was finished. The specific capacitance of 135.2 F g−1 in two-electrode system was far below of 305.5 F g−1 in three-electrode system, and the specific capacitance increased as cycle number of GCD increased, which indicates that the wettability of G/GAs has increased in the charge-discharge process. The data was put in the supporting information.
The EDLCs device performances of G/GAs were examined using two HA-2 based electrode as positive and negative electrode for practical application, the results showed the specific capacitance of 135.2 F g−1, good cycle stability, and increased wettability after 10 000 cycles at 10 A g−1.
Supporting information
Glucose/Graphene-Based Aerogels for Gas Adsorption and Electric Double Layer Capacitors
Kang-Kai Liu 1,†, Biao Jin 1,†, Long-Yue Meng 2,*
1 Department of Chemistry, Yanbian University, Yanji, 133002, China
2 Department of Polymer Materials and Engineering, Department of Chemistry, MOE Key Laboratory of Natural Resources of the Changbai Mountain and Functional Molecules, Yanbian University, Park Road 977, Yanji 133002, Jilin Province, PR China
† These authors contributed equally to this work.
* Correspondence: lymeng@ybu.edu.cn; Tel.: +82-32-876-7234; Fax: +82-32-867-5604
Supplementary Figures
Figure S1. CV curves of HA-2 (4 mg mL−1 glucose & 800 °C CO2 activation) at different scan rates.
Figure S2. GCD curves of HA-2 (4 mg mL−1 glucose & 800 °C CO2 activation) at current densities.
Figure S3. EIS curves of HA-2 (4 mg mL−1 glucose & 800 °C CO2 activation) after 1th cycle and 10 000th cycle.
Figure S4. Cyclic stability of HA-2 (4 mg mL−1 glucose & 800 °C CO2 activation).
Supplementary Discussion
To ensure the EDLCs device performances of the G/GAs based electrode, a symmetrical electrochemical capacitor device based on two G/GAs based electrode, as positive and negative electrodes, was measured in 6 M KOH. The CV curves of the device (Figure S1) keep more standard rectangular shapes than the single electrode, which exhibits the good stability of the charge transportation process. The triangles from GCD curves (Figure S2) indicate that the G/GAs based electrode exists the certain internal resistance, which results in the no linear dependence on the applied potential during the charging process, and the specific capacitance of 135.2 F g−1 was obtained at 1 A g−1. The Nyquist plots of HA-2 (Figure S3) presented no semicircle in the high frequency region, but after 10 000 cycles, the line is closer to 45° than before, which indicates the mixed control between charge transfer and diffusion processes. Furthermore, according to the specific capacitance retention after 10 000 cycles at 10 A g−1 in Figure S4, the wettability of G/GAs based electrode has increased in the charge-discharge process.
Please see the attachmenet.

Round 3
Reviewer 2 Report
Accept in present form